# Status Estimation and In-Process Connection of Kanbans Using BLE Beacons and LPWA Network to Implement Intra-Traceability for the Kanban System

**DOI:** 10.3390/s21155038

**Published:** 2021-07-25

**Authors:** Kosuke Shima, Masahiro Yamaguchi, Takumi Yoshida, Takanobu Otsuka

**Affiliations:** Nagoya Institute of Technology, Gokiso-cho, Showa-ku, Nagoya 466-8555, Japan; yamaguchi.masahiro@otsukalab.nitech.ac.jp (M.Y.); yoshida.takumi@otsukalab.nitech.ac.jp (T.Y.); otsuka.takanobu@nitech.ac.jp (T.O.)

**Keywords:** traceability, BLE beacon, LPWA network, Kanban system, IoT, IIoT

## Abstract

IoT-based measurement systems for manufacturing have been widely implemented. As components that can be implemented at low cost, BLE beacons have been used in several systems developed in previous research. In this work, we focus on the Kanban system, which is a measure used in manufacturing strategy. The Kanban system emphasizes inventory management and is used to produce only required amounts. In the Kanban system, the Kanban cards are rotated through the factory along with the products, and when the products change to a different process route, the Kanban card is removed from the products and the products are assigned to another Kanban. For this reason, a single Kanban cannot trace products from plan to completion. In this work, we propose a system that uses a Bluetooth low energy (BLE) beacon to connect Kanbans in different routes but assigned to the same products. The proposed method estimates the beacon status of whether the Kanban is inside or outside a postbox, which can then be computed by a micro controller at low computational cost. In addition, the system connects the Kanbans using the beacons as paired connection targets. In an experiment, we confirmed that the system connected 70% of the beacons accurately. We also confirmed that the system could connect the Kanbans at a small implementation cost.

## 1. Introduction

In recent years, the Internet of Things (IoT) has become a widely discussed topic. IoT is expected to enhance our health maintenance and facilities in daily life, to lead to greater efficiencies, and to find anomalies in industrial processes, among other benefits. Industrial IoT (IIoT) focuses on enhancing efficiency in work processing, intralogistics, and inventory management. Industry 4.0 has rapidly populated in recently ears. It connects products, processes, people, and physical systems more efficiently [1]. Industry 4.0 has become a main operator which is changing the manufacturing system. In addition, IoT has become one of the most rapidly growing fields in Industry 4.0.

In the industrial process, manufactured products may contain a defective part. We assume that manufacturers want to know when and at which process the defective product is output. Traceability is an important feature for determining when/which defects occur and for reducing the occurrence probability of defects. However, the introduction of a huge system for traceability is difficult for small- and medium-sized manufacturers because it requires significant cost and specific actions to implement it.

In this work, we focus on providing a product-tracing system that can be introduced at low cost and that does not require workers to take specific actions. This system consists of Bluetooth low energy (BLE) beacons and a low power wide area (LPWA) network. The BLE beacon is a very small and low-cost component that transmits Bluetooth waves at predetermined intervals. The LPWA network makes it easy to implement a network that communicates between edge devices and other devices. In addition, we focus on the Kanban system, which is a manufacturing system that orders products and provides them in only the required amounts.

The Kanban system is a well-known industrial system that focuses on just-in-time supply of only the required amount. By producing only the required amounts, the Kanban system not only reduces defective inventory and relieves pressure on the production line due to surplus production, but also contributes to the reduction of waste from production. The Kanban is a product’s order sheet as well as its shipping sheet. The Kanban is attached to pre-processed products when the products are required and their quantity is decided, and it is removed when the required products are delivered. The Kanban system is very efficient for reducing dead stock and inventory management, specifically by indicating how many products are required and processed.

Each Kanban is used for only one process, and the other processes use other Kanbans. For example, we assume a set of products that will be used at process 1 and process 2. The set is attached with the Kanban of process 1 when it is in process 1. The set removes the Kanban just before the first process is finished and then transfers to process 2, when the set becomes attached to the Kanban of process 2. In the Kanban system, the Kanbans are physical cards, so there is no digital log when the Kanbans are attached and removed from products. For this reason, the relationship among the Kanbans in the various processes is unknown, so we cannot trace the process in which the product defect occurs.

In the field of traceability, various studies focused on blockchain technology. The blockchain is used to manage whole of supply chains by connection of transactions. The transactions are information, such as a products, and were transferred to another process, a product was processed in a machine, a product was sold by retails, and so on. The blockchain enables us to trace a product when and where it was manufactured securely and reliably. However, the blockchain requires transactions for each section of the products manufactured, but the traditional Kanban system does not have digital information of transactions. Some studies implemented e-Kanban for the traditional Kanban system, however existing e-Kanban used RFID, QR, or bar code. The RFID is expensive to implement, and in addition, the QR and bar code require particular operation which may cause human error.

In this paper, we propose a system that estimates the status of the Kanbans and connects the Kanbans in the various processes to each other. Our system divides the process into five steps: the Kanbans wait to be attached to products, they are attached to products, they wait to be processed with products as stock, they remain with products until nearly processed, and finally they are removed from the products. The system attaches the BLE beacon unit to each Kanban and estimates the states of the Kanbans at the base-station using RSSI values of Bluetooth. The base-station estimates the Kanban’s state and sends it to a gateway by an LPWA network. The gateway sends the states to a server using Ethernet, and then the server connects the Kanbans among all of the processes. The proposed system is designed as inexpensive and there are no specific operations to implement for small- and medium-enterprises.

In an experiment under a test environment, we transferred the beacons in two routes and measured the RSSI values. Then we connected these beacons transferred in different routes. We correctly connected the beacons at over 70% accuracy.

## 2. Related Studies

Marini et al. developed a sensor device which measures radiological characterization of stone blocks [2]. The developed device can connect to sensor network using LoRa communication; in addition the base station of the network collects posts from the sensor devices and sends data to the database using Ethernet. Boyes et al. redefined IIoT definition and proposed a framework of the IIoT components [3]. The framework clarified IIoT components as a set of related schemes. Ding et al. clarified knowledge of IIoT, Industrial, and Projects using the ontology technique [4]. They revealed complex and huge manufacturing processes and components of recent projects.

### 2.1. Traceability

In the research on traceability, Wisanmongkol proposed a system that authenticates cars automatically using RFID [5]. RFIDs require large costs, but automatic car authentication is important for implementing traceability between factories. Shih et al. proposed a distributed traceability model for complex sequences of processes using IOTA Tangle [6]. Cao et al. also connected real manufacturers in a blockchain traceability model using an IoT component such as RFID or EPC [7]. Takahashi et al. embossed an emblem resembling a fingerprint on products [8]. The fingerprint can identify individual products. Zhong et al. implemented their system in a smart factory using RFID [9]. Bougdira et al. organized issues of modeling concept of industrial 4.0 and implementing intelligent traceability functions [10]. In addition, they proposed a cloud-based traceability system which is also based on ontology. Urbano et al. implemented traceability system for the food industry using RFID tags and WiFi communication [11]. They used RFID to connect foods to digital data and sent data to server with WiFi.

### 2.2. Blockchain and Security

Wen et al. implemented blockchain into a supply chain management system [12]. The system was developed to allow supply chain to securely and reliably manage transitions without relying on third party systems. Figorilli et al. implemented traceability system for a wood supply chain which covers everything from standing tree to retail [13]. The system contained RFID, QR, and bar code, and it authenticated all products in processes of the supply chain. Abuhasel et al. proposed SoftMax-based deep neural network for increasing security [14]. They also discussed the secure task scheduling system. Yu et al. proposed attribute-based access control scheme for smart factories [15]. The scheme enabled us to use the traceability with blockchain and securely access to manufacturing transactions. Miehle et al. proposed a consistent system, PartChain, for the traceability [16]. The system aggregated and connected nodes of supply chain networks which have different forms of databases between each one. Liu et al. determined that the blockchain researches are mostly focused on throughput of transactions while sacrificing distribution, security, and latency, a serious issue [17]. They proposed a platform which clarifies and optimizes the ability of blockchain using Deep Reinforcement Learning.

### 2.3. Kanban System

In research on the Kanban system, Ghelichi et al. implemented an RFID tag and reader for the Kanban [18]. RFID has recently been implemented in several systems because it allows us to authenticate objects easily and concurrently. Pekarcikova et al. simulated intralogistics using e-Kanban [19]. Beuster et al. implemented smart shelves for the Kanban system. They equipped an RFID reader to a shelf and also equipped a shield built with metal onto the shelf [20]. The shelf only authenticates RFID tags which are in the shelf because RFID reader can not authenticate tags through the metal plate. Thurer et al. implemented the Kanban system for waste collection operations using e-Kanban [21]. The e-Kanban system in the work uses an RFID reader and tag. The waste collection operation is also vital logistics for manufacturing. Abbadi et al. discussed about ability of the Kanban in Industry 4.0 and usage of the Kanbans in recent years [22]. In addition they proposed the Kanban 4.0 according to progress of the Industry 4.0.

### 2.4. BLE Beacons

A lot of the research works that use BLE beacons have focused on indoor positioning systems. Jae Gu et al. proposed a method that filters raw RSSI values and estimates distance, and then the method filters this estimated distance [23]. Myugin et al. simulated distance estimation and then observed its accuracy according to the number of beacons [24]. Consequently, they confirmed that estimation error decreased with an increase in the number of beacons. Subedi et al. proposed a method that estimates distance using a WCL algorithm while filtering RSSI values [25]. Watanabe et al. focused on reducing the number of BLE beacons [26]. They proposed a method that attempts to estimate an indoor position, and if the distance estimation using RSSI fails, the method makes estimations using the reflected sound of ultrasonic waves. Lam et al. focused on distance estimation for moving objects [27]. They conducted experiments to observe accuracy when transmission frequency and moving speed changed. Torii et al. proposed a method that divides a room into cells and estimates the people who are in which cell [28]. The method uses a single BLE transmitter and a receiver for estimation. Gligoric et al. used LEDs for position estimation [29]. They used LEDs because RFID, WLAN, and ultrasonic devices increase overall system costs. Lazaro et al. compared sensors which are used in IoT and IIoT projects, such as NFC, RFID, and BLE [30]. The RFID, especially UHF RFID, is expensive for implementing readers and has low costs for implementing tags.

e-Kanban applies bar codes, QR codes, and RFID to easily manage objects, logistics, and traceability using a digital database. However, small- and medium-sized manufacturers face problems in implementing it for the following reasons:Installing readers for bar and QR codes, and especially RFID, increases overall costs.These e-Kanban systems incur high computational costs.Implementing bar codes, QR codes, and RFID requires additional training for frontline workers. There could also be factors involving human error.

Consequently, we focus on attachments that implement traceability but do not require special control or change how frontline workers use the Kanban. Additionally, we apply BLE beacons, which are frequently adopted by related studies that aim to implement systems at low cost.

## 3. Proposed System

In this section we propose a system that estimates a beacon’s state and then connects different beacons. The proposed system contains the following four components:the method is based on an LPWA network that contains base-stations and a gateway,it uses many BLE beacons to implement intra-traceability in the Kanban system,it estimates a Kanban’s status using RSSI data which are sampled by the LPWA network and BLE beacons,and it connects the Kanbans used for the same product but in different processes.

### 3.1. LPWA Network: Base Station and Gateway

Figure 1 shows a system architecture of the proposed system. BLE beacons inside/outside post for the Kanban send bluetooth signal, and base-station receive signal. The proposed method uses base-stations, a gateway, and a database. The base-station contains the following components:ESP32-WROOM-32UE can receive Bluetooth and WiFi signals in the 2.4-GHz band and send major ID, minor ID, and RSSI values of received waves to ATSAME53J20A through a UART connection,ATSAME53J20A forms data into a format that is ready to store to the database and sends formed data to ES920LR2 at a UART connection,and ES920LR2 fetches formed data from ATSAME53J20A and sends them to the gateway via the LPWA network.

The gateway sends data to a database using Ethernet when it receives data from ES920LR2.

### 3.2. Kanbans and BLE Beacons

The proposed method attaches the BLE beacon to the Kanban (Figure 2). It uses FCS1301 as the BLE beacon. This beacon enables us to set UUID, Major ID, Minor ID, TxPower, and Interval. The UUID is the identification number of the beacons, so all of the beacons have different UUIDs. The Major ID identifies groups of beacons; if the beacons have different roles and move in the same area, the Major ID is efficiently used to group the beacons according to the roles. The Minor ID identifies the beacons that are in the same group. TxPower is a parameter of the strength of the Bluetooth wave transmitted by the beacon. The Interval value represents the set-up time needed for transmitting the beacons.

### 3.3. Status Estimation Method

#### 3.3.1. Assumed Flows of the Kanbans as Estimation Target

Here we present an assumed flow of the products and their Kanbans (Figure 3). This flow was modeled with the assistance of comments given by a collaborating company. Figure 3 shows two flows, where route 1 is a pre-pressing process and route 2 is a post-pressing process. The products are transferred and processed in the following sequence:When new products outputted from process 0 are attached to the Kanban (with the beacon), they are transferred to the waiting stock.The products are transferred near a machine used in Process 1, then the Kanban is removed when the products go to process 1.The products are transferred to the next route, and the Kanban is placed into the postbox.The Kanbans in the postbox are transferred to an area to wait for the new products outputted from process 0.

The above procedure is also used for route 2.

In this way, products are transferred along routes through the processes, and the Kanbans are rotated within a single route. Here, there is no information about the combinations of Kanbans that are in different routes but have been attached to the same products, since the products that removed the Kanban at route 1 have no Kanbans until new ones are attached in route 2. Therefore, we define the statuses of the Kanbans (and the beacons) and propose a method that estimates the status.

#### 3.3.2. Kanban Status

We atomize a section of just before and after process 1 based on a suggestion from a collaborating company (Figure 4) and define a sequence around process 1 as follows:The Kanban that is removed from the products is directly shot into the used postbox.Before the products are outputted from process 1, the Kanbans that will be mounted to the products wait at the pre-use postbox.After the products are outputted, one of the Kanbans that are in the pre-use postbox is immediately attached to the products.

Accordingly, the atomized sequence indicates that there is a used postbox in route 1 and a pre-use postbox in route 2, in addition the Kanban of route 1 is removed and the Kanban of route 2 is attached to the products in a continuous operation.

Then we defined the Kanban’s status as the following two states for each route:Plus state means that the Kanbans are in the postbox.Minus state means that the Kanbans are not in the postbox but attached to the products.

The proposed system sets a base-station above each postbox (Figure 5 and Figure 6). Each figure is fashioned after the real postboxes used at the collaborating company. If the postboxes are made of metal, the Bluetooth wave transmitted from a Kanban in the postbox are amplified by reflection in the box. In addition, waves transmitted from outside of the postboxes attenuate according to the distance between the base-station and the beacons. For this reason, the proposed system focuses on the RSSI values, which are set to be different between the plus and minus states.

#### 3.3.3. Status-Estimation Method

The status-estimation method is a light-duty process because we assume that the method is driven on edge devices. The method is performed in the four steps shown in Figure 7 and explained as follows:The method requires a teacher dataset. The teacher dataset is sampled with the beacons that stay at a predetermined area and do not move. The data have a fetched timestamp, a Major ID, a Minor ID, and RSSI values and are sampled when the base-station receives the wave from the beacons. The method calculates distances between all of the data sample points using RSSI values and clusters all samples by hierarchical clustering using the Ward method.After clustering, the method calculates mean and variance for each cluster and acquires the normal distribution for each cluster.It decides five thresholds according to the confidence interval. All of the thresholds have approximate RSSI values. The values are 10% of the normal distribution of the plus state, the midpoint between 1% of the plus state and 99% of the minus state, 90% of the minus state, 10% of the minus state, and 1% of the minus state. We denote these thresholds as threshold 1 to threshold 5, respectively.Using test data, it marks flags for the Minor ID of the beacons while checking all samples by temporal sequence. The flags have two categories, plus flag and minus flag. The method marks the flags according to the above thresholds: It marks three plus flags to the Minor ID *A* when the Minus ID of a sample is *A* and the RSSI value of the sample is larger or equal to threshold 1, a plus flag when RSSI is in [threshold 2, threshold 1), a minus flag in [threshold 3, threshold 2), three minus flags in [threshold 4, threshold 3), a minus flag in [threshold 5, threshold 4), and marks no flag when RSSI is lower than threshold 5.In addition, it records a timestamp when the first one of the flags is marked for each Minor ID.When the last 5 flags of the Minor ID *A* are the same flag, 5 plus flags or 5 minus flags, then the method decides the state of the corresponding beacon and stops observing the Minor ID *A*.

The used Kanbans in the used postbox are transferred to the pre-used postbox at a predetermined time everyday. Then, the method restarts, observing the Minor ID when the timestamp of the sample that is currently being checked is past a predetermined timestamp. The method outputs a table that contains a timestamp when the beacon’s state changes, the Minor ID of the beacon, and the state of either plus or minus.

The status estimation method requires teacher data that is sampled under the condition that the beacons are still at the predetermined area. We sampled a static state dataset using the following procedure:first we grouped 8 beacons into 4 groups,next we positioned the beacon groups into 4 predetermined points (Figure 8, with point 3 outside of this room),then we rotated each group to the other points every 24 h.

Figure 8 shows the points of the beacon groups, which are the postbox, point 1, point 2, and outside of this room. The distance from the postbox is about 3 m at point 2 and about 5 m at point 3. We sampled data for 12 days and sampled 97,437 samples. Figure 9 shows a histogram of the RSSI values for all data samples. In Figure 9, the horizontal axis is RSSI value and the vertical axis is probability. The method divides data samples into two clusters using hierarchical clustering. The red line and blue line in Figure 9 are normal distributions calculated with mean and variance according to each cluster. The normal distributions show joint probability of the plus and minus states according to the RSSI measure. For these distributions, the method determines the thresholds 1 to 5 as −40.0, −52.5, −66.0, −80.0, and −86.0, respectively.

### 3.4. Beacon-Connecting Method

Next, we propose a method that connects the beacons (and the Kanbans) using the state of the beacon and its timestamp of when the state has been estimated, acquired by the status estimation method. The connecting method is modeled like a greedy process.

This method requires the output of the status estimation method and the temporal length of the process between routes, for example, process 1, which is between route 1 and route 2 in Figure 4, as input data. We denote the set of timestamps of the input data, the set of timestamps of route 1, and that of route 2 as *T*, $T_{1}$, and $T_{2}$, respectively ($T=T_{1}\cup T_{2}$). Here, the sets $T_{1}$ and $T_{2}$ are divided using the Minor ID of the beacons. We also denote the temporal length of the process and the margin as Δt and dt, respectively. We define the following equations that acquire candidate time ranges.
$$Candidate_{1}=\Delta t-dt+t_{1}\in T_{1},$$
(1)$$Candidate_{2}=\Delta t+dt+t_{1}\in T_{1}.$$

If a timestamp $t_{2}\in T_{2}$ is in the candidate time range [$Candidate_{1}$, $Candidate_{2}$], then the connecting method connects the rows of input data that have timestamps ${}_{1}$ and $t_{2}$. The method anomalistically connects rows according to row number when the timestamps $t_{1},t_{3}\in T_{1}$ (also $t_{2},t_{4}\in T_{2}$) have completely the same value.

The method repeats the above process with incrementing margin dt of 60 s, while dt is smaller than Δt. Here, when the rows that have timestamps $t_{1}$ and $t_{2}$ are connected, the rows are no longer connected with the other rows.

### 3.5. Summary of Characteristics of Proposed System

We propose the system described above for implementing traceability at low cost and no requirement of training frontline workers. The characteristics of the proposed system are as follows:The system requires postboxes constructed of metal, but it permits estimating the status of the BLE beacons. It can estimate the status for each beacon using a single receiver. Thus, the system allows us to perform one-on-one status estimation using BLE.The status-estimation algorithm is designed to be driven on a micro controller such as ATSAME53J20A and an embedded system such as a base-station; in addition, it does not require costly computational resources such as smartphones.The system can divide status estimation (which requires a small computational cost) and the beacon connection (which requires some computational costs) into edge side and server side, respectively, since the system divides the edge side and the server side using an LPWA network. In addition, the LPWA network requires low running and implementation costs.

## 4. Experiment

### 4.1. Experimental Conditions

We conducted an experiment to measure how accurately the proposed system estimates the status and connects the beacons. Table 1 shows the experimental conditions. As experimental conditions, we used 8 BLE beacons for each route and the same settings for each beacon except for the Minor ID. We set the Major ID for all beacons to 001; we used the Major ID as a key part of the application that uses the beacons for such purposes as anomaly detection in intralogistics [31] and optimization for complement decision [32]. The Minor ID is the main identifier for the beacons in this experiment, and it is set to 131–138 for 8 beacons in route 1 and to 121–128 for 8 beacons in route 2. TxPower is 0, which is set so that the beacons transmit a Bluetooth wave in which the mean RSSI value becomes about −50 at 1 m from the ESP32-WROOM-32UE. Interval shows the duration of waiting for the beacon that is already transmitting the Bluetooth wave.

In the other setting, we used five flags for the beacon-status estimation. We assumed that the process between the routes has a fixed duration, so we set the duration of process Δt to 10 min. We set the initial value of margin dt to 1 min.

We sampled test data under the experimental environment shown in Figure 10. In reference to Figure 4, we assume point 1 in Figure 10 as the products with the Kanban close to process 1. Postbox 1 in Figure 10 is assumed to be the used postbox in Figure 4. Also as mentioned above, postbox 2 and point 2 are assumed to be the pre-use postbox and the Kanban with the products near process 1 at route 2. Then point 1 and postbox 1 are part of route 1, and postbox 2 and point 2 are part of route 2. We transferred the beacons in the following steps:We positioned beacons 131–138 on point 1 and beacons 121–128 on postbox 2. In addition, we checked the timestamp of this timing.We transferred beacon 131 from point 1 to postbox 1.We transferred beacon 120+n from postbox 2 to point 2 and beacon 131+n from point 1 to postbox 1. Here, *n* is 1 as initial, 7 as max, and incremented each time of this transfer.We transferred beacon 128 from postbox 2 to point 2.We repeated the above until *n* became 7. Here, we drove the above with 10-min intervals.

By taking the above sequences, 8 connecting targets are sampled. The connecting targets are sets of beacon 130+n and 120+i, which are transferred 10 min after beacon 130+i is transferred (*i* is 1 to 8). We repeated the above sequence 34 times and sampled 272 (34 × 8) connecting targets and 57,709 test data samples.

### 4.2. Results

#### 4.2.1. Beacon-Status Estimation

First we conducted status estimation of the beacons. Figure 11 shows the results of the order of status estimated for each route. In route 1, the vertical axis shows the Minor ID 131–138 and the horizontal axis shows the order of estimation. If all of the estimations are collected, the number of points in the graph is 272, and the points form 37 straight lines. If a line is bent like the 1st, 6th, 7th, and 9th lines and lacks points like the 9th, 10th, 13th, and 18th lines from the left, there will be estimation errors caused by the failure to receive the Bluetooth wave, lack of communications in the LPWA network, hardware resetting with continuous duty, and so on. In route 2, the vertical axis shows the Minor ID 121–128, and the others are related to route 1.

In the results, 20 (9 in route 1 and 11 in route 2) switchings of estimation orders and 17 (8 in route 1 and 9 in route 2) estimation failures occurred. We counted estimation errors by switching × 2 + failures because switching causes two connection errors for the beacon-connecting method. Therefore, the estimation errors became 57. There were 544 state changes (=272 samples × 2 routes), and the estimation errors occurred at a rate of 10.47%.

#### 4.2.2. Beacon-Connecting Results

Using the results of the beacon-status estimation, we conducted experiments to connect the beacons. The proposed system connected the 263 estimated statuses of route 2 to the 264 statuses of route 1. Table 2 shows results for the percentage of correct connections.

In the results, 70.22% of 264 statuses of route 1 were correctly connected and 9.93% were incorrectly connected. Furthermore, 17.28% of the 264 statuses of route 1 could not find the connecting target because no beacon was in the candidate time range or there was only one target already connected by another beacon. For the purpose of implementing traceability in the Kanban system, we assume not only that the percentage of not finding a target was a small value but also that incorrect connections having a small value is an important characteristic of the proposed system. In the Kanban system, we assume that confusion of connections between the Kanbans is a serious problem because it causes one to mistake a normal process for an abnormal process when tracking a defective product. For this reason, we believe it is necessary to reduce incorrectly found connections before reducing connection failures in future work.

## 5. Managerial Implications

In management of the Kanban and the BLE beacons, the proposed system does not require additional operation to implement. Frontline workers can use the system with their usual routine. However, the system requires button battery replacement for each beacon once a year.

In addition, since the only data transmitted from the beacon and stored in the server is RSSI, it does not make sense by itself and cannot be linked to the actual factory, so it is considered to be a secure system.

Compared to existing Kanbans, the proposed system requires battery replacement once a year, but instead of requiring battery replacement, transaction data can be obtained during Kanbans just by implementing it in the usual business form. In addition, compared to the existing e-Kanban, the proposed system does not need to be read by a reader like QR and bar code, and can be implemented at a lower cost than RFID. However, it should be noted that the accuracy may be affected by data loss due to increased traffic and beacon malfunction. Furthermore, BLE beacons are much cheaper to read than RFID, but RFID is cheaper in tags. Therefore, in a large-scale Kanban system owned by a large company, it is possible that the introduction cost of the system using RFID may be lower depending on the number of Kanbans.

For the above reasons, it is appropriate to introduce a system using RFID for large enterprises with large-scale systems, and to introduce the proposed system for small and medium-sized enterprises mentioned in this paper. The proposed system is considered to be effective in that it can present options for small and medium-sized companies according to the scale of the Kanban system.

## 6. Conclusions

In this paper we proposed a system that estimates the status of a Kanban and connects Kanbans between different manufacturing processes. The system is constructed with BLE beacons, base-stations, gateways, and servers. The BLE beacons are attached to the Kanbans and transmit Bluetooth waves. Next, the base-station receives the waves and transmits an LPWA connection to the gateway, which allows us to implement the system at low cost. The system has two methods: a status-estimation method and a beacon-connection method. The estimation method can divide the beacon status into being in the postbox and being outside the postbox with low computational costs, which allows us to drive the system with a micro controller. The connection method is driven in the manner of a greedy method and automatically connects the Kanbans that are in different lines but have been attached once to the same products. Furthermore, the method does not require additional operations by frontline workers.

We conducted an experiment to transfer the beacons in an assumed environment of manufacturers. From the results, the proposed system estimated about 90% of the 544 beacon-status changes, and about 70% of the beacons were correctly connected. The results show 30% connection error, which is mostly due to lost connection targets but partly to incorrect connections. We assume that incorrect connection is a more serious problem than connection loss, so we will strive to reduce incorrect connections in future work. We will continue to strive to improve the accuracy of the beacon connection with the goal of a correct answer rate of 95%.

Furthermore, in future work we will implement the proposed system in real environments. We assume that the proposed method can be extended to other targets that use boxes such as stock management, so we will also focus on implementing the system for several targets.

## Figures and Tables

**Figure 1 sensors-21-05038-f001:**
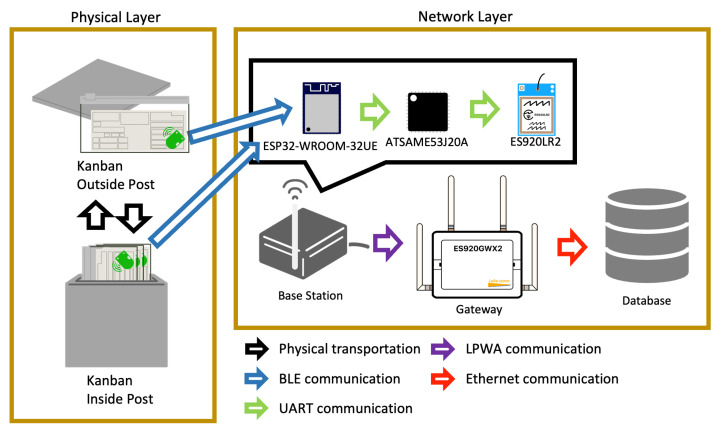
System Architecture overview of the proposal.

**Figure 2 sensors-21-05038-f002:**
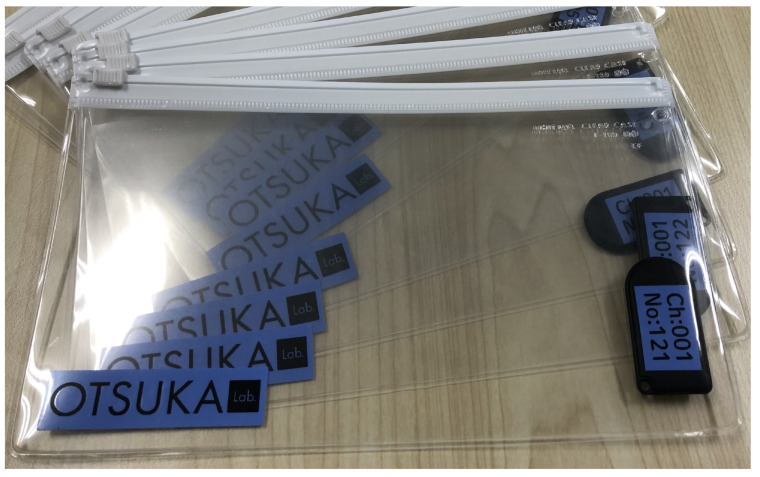
Test Kanbans and BLE beacons.

**Figure 3 sensors-21-05038-f003:**
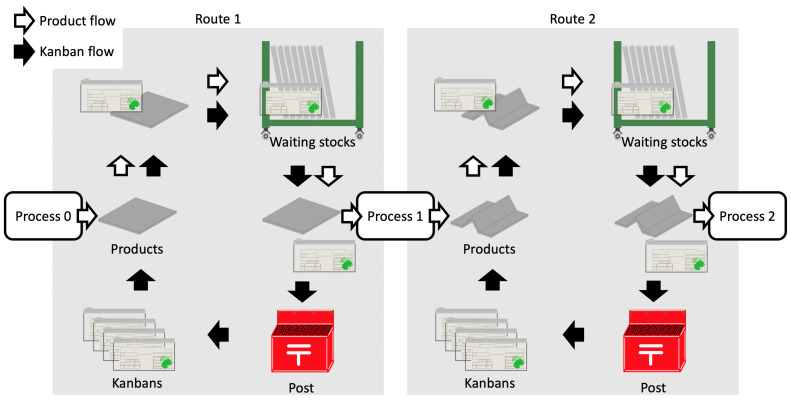
Assumed flows of the products and Kanbans.

**Figure 4 sensors-21-05038-f004:**
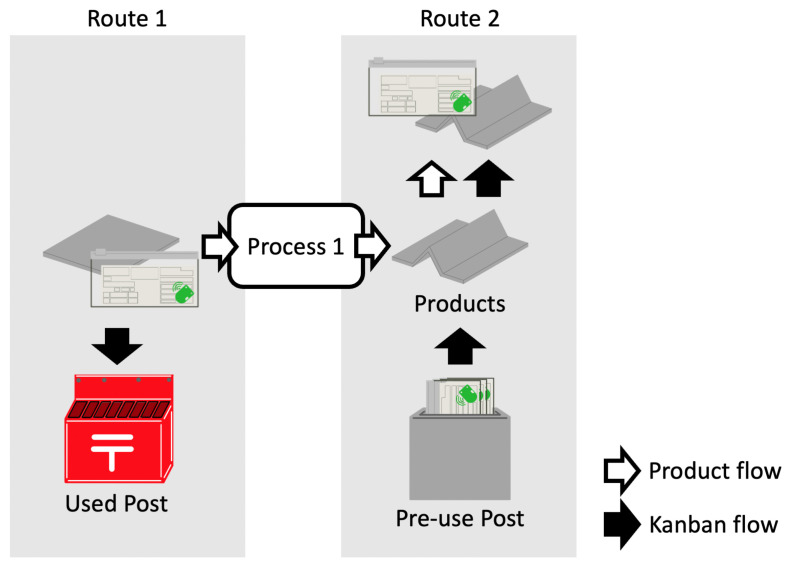
Detail of just before and after process 1.

**Figure 5 sensors-21-05038-f005:**
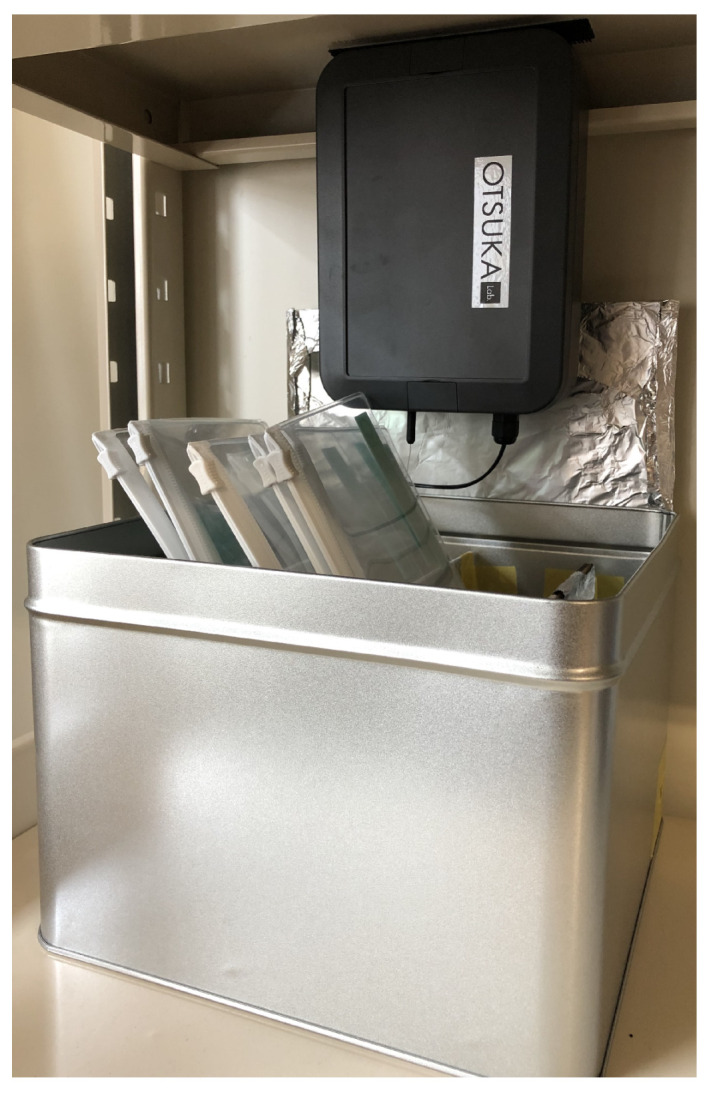
Prototype of Postbox 1 for the Kanbans.

**Figure 6 sensors-21-05038-f006:**
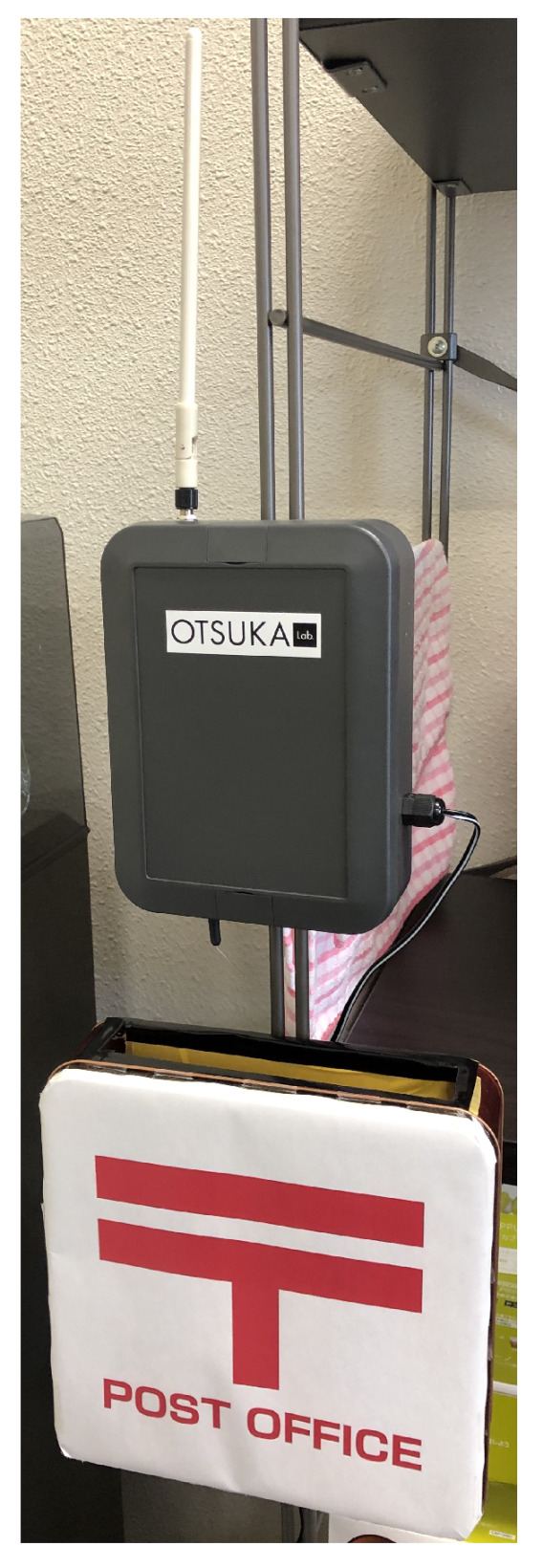
Prototype of Postbox 2 for the Kanbans.

**Figure 7 sensors-21-05038-f007:**
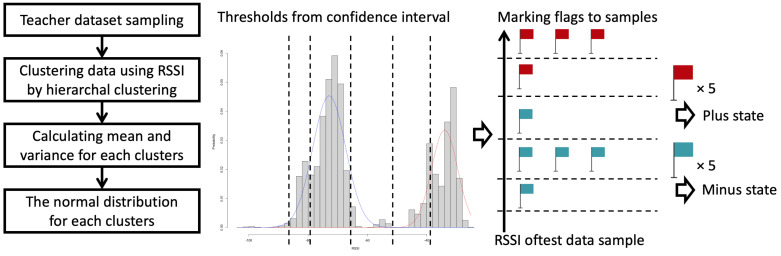
Overview of status estimation method.

**Figure 8 sensors-21-05038-f008:**
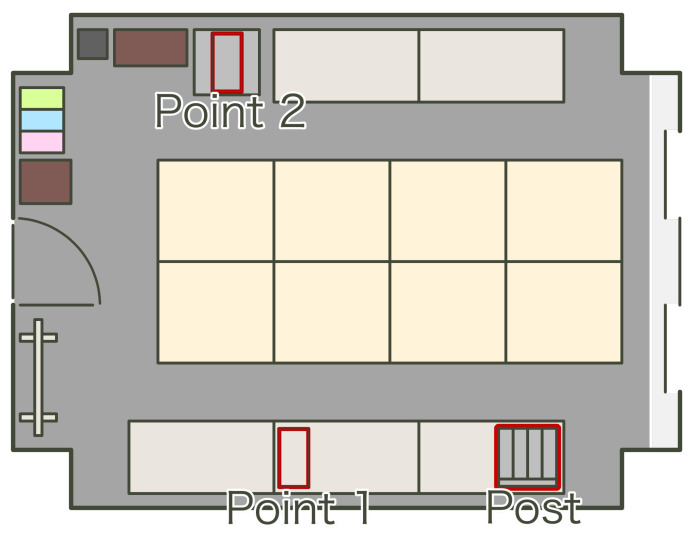
Experimental environment; in this experiment, the beacons are the points transported by rotation.

**Figure 9 sensors-21-05038-f009:**
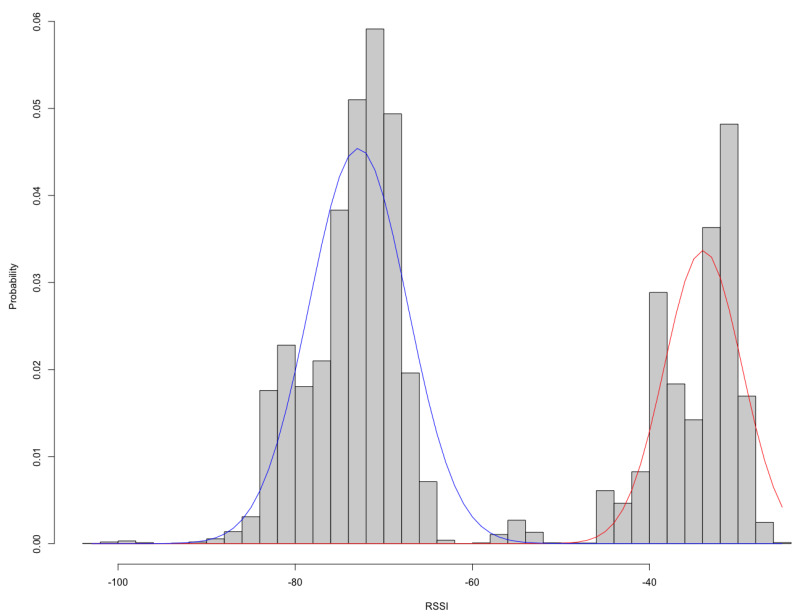
Histogram of teacher data; x axis is RSSI values and y axis is density.

**Figure 10 sensors-21-05038-f010:**
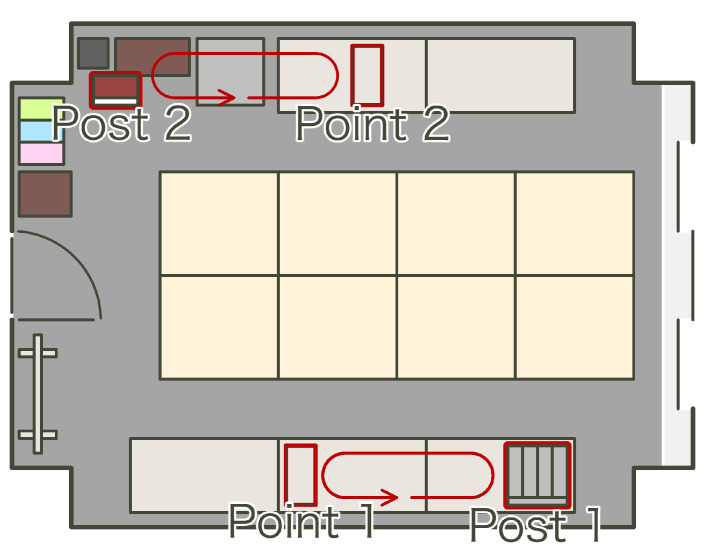
Experimental environment; in this experiment the beacons are transported one by one at predetermined intervals.

**Figure 11 sensors-21-05038-f011:**
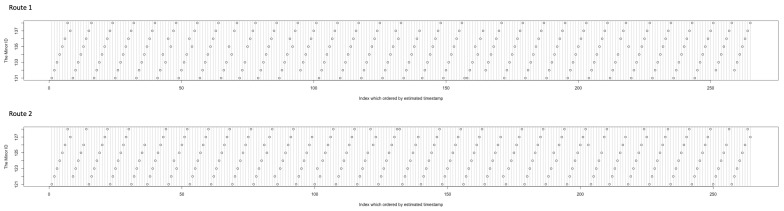
Order of beacon’s status estimated; Minor ID 131–138 (also 121–128) form 34 lines when all estimations are correct.

**Table 1 sensors-21-05038-t001:** Summary of experimental conditions.

	Route 1	Route 2	The Other Settings	
Number of beacons	8		Number of the flags	5
Major ID	001		Duration of the process Δt	10 min
Minor ID	131 to 138	121 to 128	Initial setting for the margin dt	1 min
TxPower	0		Number of test data Samples	55,709
Interval	3000 ms		Number of connecting targets	272

**Table 2 sensors-21-05038-t002:** Accuracy of connecting beacons.

Correct Connecting	Incorrect Connection	Connecting Target Is Not Found	Status of Route 1 Is Not Found
70.22%	9.93%	17.28%	2.57%

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
