# Peer review of "Status Estimation and In-Process Connection of Kanbans Using BLE Beacons and LPWA Network to Implement Intra-Traceability for the Kanban System"

_sensors, 2021, doi:10.3390/s21155038_

Round 1

Reviewer 1 Report

The manuscript, titled as “Status Estimation and In-process Connection of the Kanbans using BLE Beacons and LPWA Network to Implement Intra-traceability for the Kanban System”, presents an interesting and practical work to enhance the functionality of Kanban which is well-known in the area of industrial engineering.  Overall, the value of Industrial IoT in the context of Kanban are revealed, but the whole manuscript should be proofread in the revision. The comments for further improvements are listed as follows:

  • Several grammatical mistakes are found, and the choice of words and sentence structures are weird. Authors should proofread the work carefully in the revision before the next submission.
  • In Section 1, authors should elaborate more the motivation of this study. Why do you consider IoT technologies in the Kanban? Are there any pitfalls/weakness of the typical Kanban in the modern business environment? Is this study solely developed for small and medium enterprises (SMEs)?
  • In Section 2, authors should review the recent development of IoT/IIoT technologies, instead of describing the technical backgrounds. You may obtain some insights from the latest review studies (see below as suggestions):
  1. Smart Sensors and Industrial IoT (IIoT): A Driver of the Growth of Industry 4.0. In Smart Sensors for Industrial Internet of Things (pp. 37-49). Springer, Cham. (2021).
  2. Exploring the intellectual cores of the blockchain–Internet of Things (BIoT). Journal of Enterprise Information Management. (2021).
  3. Using intelligent ontology technology to extract knowledge from successful project in IoT enterprise systems. Enterprise Information Systems. (2021).
  • In Section 3, authors should provide a service-oriented architecture for your proposed system, which is important to graphically illustrate the IIoT deployment.
  • Also, justifications for the hardware selection should be provided.
  • In Section 4.2, the results of connection accuracy, i.e. 70.22%, seems not good enough. More experiments should be conducted to further validate the results, or authors may consider to enhance the proposed system.
  • The practicality and contribution of this study are proven, which is novel in the context of e-Kanban. A section of managerial implications should be added before Conclusion.

Author Response

We are very grateful for your many appropriate suggestions.
We would like to reply to the pointed out content.

For the sake of clarity, the points pointed out are shown in blue, the answers are shown in black, and the corrections and additions to the manuscript are shown in green.

Several grammatical mistakes are found, and the choice of words and sentence structures are weird. Authors should proofread the work carefully in the revision before the next submission.

In order to make the English expression correct, the manuscript was proofread in English and the entire sentence was corrected based on the result.

In Section 1, authors should elaborate more the motivation of this study. Why do you consider IoT technologies in the Kanban? Are there any pitfalls/weakness of the typical Kanban in the modern business environment?

One of the weaknesses of conventional Kanban systems is that the process using a physical Kanban card does not provide the transaction data required for blockchain, which has been widely studied in recent years.
Therefore, there are studies dealing with kanban as digital data, but the main ones are RFID, which is expensive to install a reader, and QR and barcode, which may cause human error because special operations are required.
Therefore, a BLE Kanban that can be implemented at low cost is required.
To clarify the above, we have added the following paragraphs to Section 1.

<Added Paragraph>

"In the field of traceability, various studies focused on blockchain technology. 
The blockchain is used to manage whole of supply chains by connection of transactions. 
The transactions are information such as a product was transferred to another process, a product was processed in a machine, a product was sold by retails, and so on.
The blockchain enables us to trace a product when and where it have be manufactured securely and reliably. 
However, the blockchain requires transactions for each section of that the products are manufactured, but the traditional Kanban system do not have digital information of transactions. 
Some studies implemented e-Kanban for the traditional Kanban system however existing e-Kanban used RFID, QR, or bar cord. 
The RFID is expensive to implement reader, in addition, the QR and bar cord require particular operation which may course human error."

 Is this study solely developed for small and medium enterprises (SMEs)?

Yes, the method in this paper is a system developed for SMEs.
The following text has been added to Section 1 to be explicit.

"The proposed system is designed as inexpensive and no specific operation, to implement for small- and medium-enterprises."

In Section 2, authors should review the recent development of IoT/IIoT technologies, instead of describing the technical backgrounds. You may obtain some insights from the latest review studies (see below as suggestions):

  1. Smart Sensors and Industrial IoT (IIoT): A Driver of the Growth of Industry 4.0. In Smart Sensors for Industrial Internet of Things (pp. 37-49). Springer, Cham. (2021).
  2. Exploring the intellectual cores of the blockchain–Internet of Things (BIoT). Journal of Enterprise Information Management. (2021).
  3. Using intelligent ontology technology to extract knowledge from successful project in IoT enterprise systems. Enterprise Information Systems. (2021).

By adding 15 papers including the recommended manuscripts and dividing the structure of Section 2 by classification, we have organized it so that you can get a bird's-eye view of the recent research situation.

In Section 3, authors should provide a service-oriented architecture for your proposed system, which is important to graphically illustrate the IIoT deployment.

We are delighted for your appropriate indicate. Figure 1 has been changed to show the entire proposed system. In addition, referring to a number of surveyed papers, we divided the system architecture shown in the figure into a physical layer and a network layer.

Also, justifications for the hardware selection should be provided.

Regarding the point that we used BLE instead of RFID and QR, which have been often taken up in conventional research, we added a survey paper comparing BLE, RFID, NFC, and UHF RFID in Section 2, BLE is the least expensive, and we have decided that it can be implemented without requiring special operations from humans.

In Section 4.2, the results of connection accuracy, i.e. 70.22%, seems not good enough. More experiments should be conducted to further validate the results, or authors may consider to enhance the proposed system.

Regarding the accuracy of beacon connection, we still feel that it is necessary to improve the accuracy. However, in this experiment, the purpose was to confirm that the beacon could be connected using only the time stamp and RSSI as inputs, so the accuracy will be determined in the future.

In response to this, we have added the following text to Section 5 to clarify future issues.

"We will continue to strive to improve the accuracy of the beacon connection with the goal of a correct answer rate of 95%."

The practicality and contribution of this study are proven, which is novel in the context of e-Kanban. A section of managerial implications should be added before Conclusion.

Regarding the management of beacons used in the system proposed in this paper, we have added the following in the form of adding a section.

"In management of the Kanban and the BLE beacons, the proposed system don't require additional operation to implement. Frontline workers can use the system with usual routine.
However, the system requires button battery replacement for each beacons once a year.

In addition, since the only data transmitted from the beacon and stored in the server is RSSI, it does not make sense by itself and cannot be linked to the actual factory, so it is considered to be a secure system."

Reviewer 2 Report

Dear authors,

Your research is very interesting and actual for improving the manufacturing process.

  in figure 8 "Calcurating mean and variance" must be "Calculating mean and variance"maybe

the description of Figure 9 has "histgram" instead of "histogram"

 Congratulation on your work!

Author Response

We are very grateful for the peer review.

I would like to  answer the points you pointed out.

For the sake of clarity, the points pointed out are shown in blue and the answers are shown in black.

  in figure 8 "Calcurating mean and variance" must be "Calculating mean and variance"maybe

the description of Figure 9 has "histgram" instead of "histogram"

As you pointed out, there was a mistake in the spelling of English words, so I corrected it.

Reviewer 3 Report

A system for estimating the status of Kanban systems is proposed. Here, the Authors provide an architecture comprising beacons, base-station, gateway and server to define a system which estimates the  status of Kanban systems between manufacturing processes.

The context presented is interesting. However, at the time of reading, the manuscript is not in a publishable status yet. Here are some aspects the Authors should address:

  • Introduction should be completely rewritten in order to better highlight the contributions of the work. Kanban system should be explained clearer and the rationale behind the proposed approach should be stated.
  • Related studies should be improved, in the sense that the Authors should position their proposed approach within the related ones.
  • Several typos and grammar errors are present. The Author should completely revise the manuscript and fix those.

Author Response

We are very grateful for your polite and kind suggestions.
We would like to reply to the points pointed out.

For the sake of clarity, the points pointed out are shown in blue, the answers are shown in black, and the corrections and additions to the manuscript are shown in green.

Introduction should be completely rewritten in order to better highlight the contributions of the work. Kanban system should be explained clearer and the rationale behind the proposed approach should be stated.

We have clarified the requirements and purpose by adding an explanation about the Kanban system and then describing the problems that the current Kanban system has and the developed e-Kanban.
We also paid attention to blockchain, which has been popular in recent research on traceability, and added a paragraph stating that e-kanban is necessary for the introduction of blockchain.

Added and Modified Paragraphs

"The Kanban system is a well-known industrial system that focuses on just-in-time supply of only the required amount.
By producing only the required amounts, the Kanban system not only reduces defective inventory and relieves pressure on the production line due to surplus production, but also contributes to the reduction of waste from production.
The Kanban is a product's order sheet as well as its shipping sheet.
The Kanban is attached to pre-processed products when the products are required and their quantity is decided, and it is removed when the required products are delivered.
The Kanban system is very efficient for reducing dead stock and inventory management, specifically by indicating how many products are required and processed."

"In the field of traceability, various studies focused on blockchain technology.
The blockchain is used to manage whole of supply chains by connection of transactions.
The transactions are information such as a product was transferred to another process, a product was processed in a machine, a product was sold by retails, and so on.
The blockchain enables us to trace a product when and where it have be manufactured securely and reliably.
However, the blockchain requires transactions for each section of that the products are manufactured, but the traditional Kanban system do not have digital information of transactions.
Some studies implemented e-Kanban for the traditional Kanban system however existing e-Kanban used RFID, QR, or bar cord.
The RFID is expensive to implement reader, in addition, the QR and bar cord require particular operation which may course human error."

Related studies should be improved, in the sense that the Authors should position their proposed approach within the related ones.

Regarding Section 2, we additionally cite 15 papers published in recent years and rewrite them so that we can get a bird's-eye view of recent research background by classifying them into traceability, blockchain, kanban, and BLE beacon.

Several typos and grammar errors are present. The Author should completely revise the manuscript and fix those.

In order to review the English grammar as a whole, we proofread the manuscript and then revised the entire manuscript.

Round 2

Reviewer 1 Report

The authors have revised the manuscript according to my comments appropriately. The quality of this paper has been improved a lot, which should be publishable in this journal soon. In my opinion, the managerial implications are relatively short and less-informative. I suggest that the authors may explain more about the pros & cons obtained by the deployment of the proposed system, and any differences/improvements compared with existing Kanban/e-Kanban. In addition, operational and social impacts from the development of the proposed system can be discussed. Instead of discussing the technical side of this work, the authors may consider to explain more about the contributions in the industry and society. 

Author Response

Thank you for your peer review.
The following paragraphs have been added to Section 5 to clarify the weaknesses and significance of the proposed system.

"Compared to existing Kanbans, the proposed system requires battery replacement once a year, but instead of requiring battery replacement, transaction data can be obtained during Kanbans just by implementing it in the usual business form. In addition, compared to the existing e-Kanban, the proposed system does not need to be read by a reader like QR and barcode, and can be implemented at a lower cost than RFID. However, it should be noted that the accuracy may be affected by data loss due to increased traffic and beacon malfunction. Furthermore, BLE beacons are much cheaper to read than RFID, but RFID is cheaper in tags. Therefore, in a large-scale Kanban system owned by a large company, it is possible that the introduction cost of the system using RFID may be lower depending on the number of Kanbans.

For the above reasons, it is appropriate to introduce a system using RFID for large enterprises with large-scale systems, and to introduce the proposed system for small and medium-sized enterprises mentioned in this paper. The proposed system is considered to be effective in that it can present options for small and medium-sized companies according to the scale of the Kanban system."

Reviewer 3 Report

The Authors successfully addressed my concerns.

Author Response

Thank you for the peer review.
I will continue to concentrate on writing clear and easy-to-understand treatises.